# TLR agonists polarize interferon responses in conjunction with dendritic cell vaccination in malignant glioma: a randomized phase II Trial

Richard G. Everson [1,2,9], Willy Hugo [2,3,4,9], Lu Sun[1], Joseph Antonios [1], Alexander Lee [1,5], Lizhong Ding[3], Melissa Bu [3], Sara Khattab [1], Carolina Chavez[5], Emma Billingslea-Yoon[1], Andres Salazar[6], Benjamin M. Ellingson[2,7], Timothy F. Cloughesy [5,8], Linda M. Liau [1,2] ✉ & Robert M. Prins [1,2,4,5] ✉

In this randomized phase II clinical trial, we evaluated the effectiveness of adding the TLR agonists, poly-ICLC or resiquimod, to autologous tumor lysate-pulsed dendritic cell (ATL-DC) vaccination in patients with newly-diagnosed or recurrent WHO Grade III-IV malignant gliomas. The primary endpoints were to assess the most effective combination of vaccine and adjuvant in order to enhance the immune potency, along with safety. The combination of ATL-DC vaccination and TLR agonist was safe and found to enhance systemic immune responses, as indicated by increased interferon gene expression and changes in immune cell activation. Specifically, PD-1 expression increases on CD4+ T-cells, while CD38 and CD39 expression are reduced on CD8+ T cells, alongside an increase in monocytes. Poly-ICLC treatment amplifies the induction of interferon-induced genes in monocytes and T lymphocytes. Patients that exhibit higher interferon response gene expression demonstrate prolonged survival and delayed disease progression. These findings suggest that combining ATL-DC with poly-ICLC can induce a polarized interferon response in circulating monocytes and CD8+ T cells, which may represent an important blood biomarker for immunotherapy in this patient population. Trial Registration: ClinicalTrials.gov Identifier: NCT01204684.

There have been significant advances in our genetic and immunologic understanding of primary brain tumors, such as malignant gliomas. Yet, it has still proven difficult to improve long-term outcomes in patients using standard-of-care therapies[1]. We and others have demonstrated that autologous tumor lysate (ATL) dendritic cell (DC) vaccination can induce local and systemic anti-tumor immune responses in malignant glioma patients, and clinical trials have suggested that this may improve survival in this deadly condition[2-6]. However, variable response rates in

cancer immunotherapy trials have prompted the search for strategies to enhance cancer vaccine potency. In particular, agonists of a family of pattern-recognition receptors (PRR) called Toll-like receptors (TLR)[7-10], which appear capable of activating of antigen-presenting (i.e., dendritic) cells, enhancing T-cell priming, and decreasing myeloid-derived suppressor cells (MDSC), are rational candidates for use in combination with ATL-DC vaccination to potentially enhance the antitumor immune response[10,11].

TLR3 is an intracellular PRR that recognizes double-stranded RNA (dsRNA), usually associated with viral infection, and induces high levels of IFN-α/β and pro-inflammatory cytokines when activated. TLR-3 is predominantly expressed by macrophages, plasmacytoid DC and myeloid DC[12,13], as well as microglial cells[14,15]. It has also been shown that astrocytes[16–18] and malignant gliomas[19] also respond similarly to TLR3-induced signaling. Polyinosinic acid-polycytidylic acid stabilized with polylysine (poly-ICLC) is a multi-dimensional synthetic dsRNA analogue and viral mimic that signals via TLR3, MDA5 and other dsRNA-dependent PRR signaling, induces type I-II IFNs[20,21], promotes the infiltration of effector T cells in pre-clinical glioma models[22], and upregulates genes associated with chemokine activity, T-cell activation, and antigen presentation[23]. Poly-ICLC has been tested as a single-agent therapeutic for multiple malignancies[24], including malignant glioma patients[25], in whom it has demonstrated adequate safety, but limited survival benefit in combination with standard-of-care therapies[26].

Similarly, TLR7 and TLR8 are other intracellular PRRs that recognize single-stranded RNA (ssRNA), which subsequently induces proinflammatory cytokines, chemokines, and type I interferons (IFNs)[27]. In pre-clinical work, we previously demonstrated that DC injected into imiquimod (TLR7 agonist)-pre-treated sites acquired lymph node migratory capacity and enhanced T-cell priming[28]. Our early phase clinical trials demonstrated that DC vaccination with adjuvant topical imiquimod, a TLR-7 agonist, was safe and feasible in glioblastoma patients[3]. Resiquimod is a newer imidazoquinoline agonist that shows enhanced transdermal delivery, activates TLR7/8 to enhance T-cell responses and TH1-type cytokine secretion by DC[29–32], and may have greater potency as an immune modulator.

In this study, we report the long-term results of 23 malignant glioma patients enrolled in a phase II randomized clinical trial where patients were randomized to receive Poly-ICLC, Resiquimod or a placebo in addition to ATL-DC. The trial was designed to evaluate the immunologic effects of the addition of the TLR agonists and compare the safety, immune responses, and potential efficacy. Post-hoc analysis using cytometry by time-of-flight (CyTOF) and bulk and single-cell RNA sequencing (scRNAseq) technologies were used to detect the cellular and molecular immune signatures from peripheral blood mononuclear cells (PBMCs) pre- and post-treatment.

## Results

### Patient characteristics and safety

A total of 23 patients with resection-eligible WHO grade III or IV glioma were enrolled and randomized between September 2010 and August 2014. All patients received ATL-DC vaccination as an initial series of 3 biweekly bilateral upper extremity injections of 2.5x10e6 ATL-DCs followed by up to 7 booster injections at 4-month intervals. Randomization allocated nine into the adjuvant TLR-7/8 agonist (resiquimod, 0.2% gel, 3M, applied to ATL-DC injection site days 0, 2, 4 post-DC injection) group, nine into the adjuvant TLR-3 agonist (poly-ICLC, 20 mcg/kg IM, Oncovir, upper extremity, at time of DC injection) group, and five to the adjuvant placebo arm where patients received either carrier gel without resiquimod or IM saline injection. (Fig. 1A, Supplementary Fig. 1). All patients were followed for clinical evaluations, toxicity, survival, imaging changes, as well as in-depth systemic immune monitoring. Baseline patient characteristics are presented and segregated by treatment group in Table 1 (see also Supplementary Data 1). The median age was 45.3 (range 26.2–72.8) years, and 57% of the enrolled patients were male. Patients were enrolled prior to the 2016 update to the WHO classification of central nervous system tumors; 65% (n = 15) had histopathological diagnoses of WHO Grade IV glioblastoma (now consistent with IDH wild-type glioblastoma), while 35% (n = 8) of the patients were WHO Grade III (all of which would now classify as IDH-mutant astrocytoma or oligodendroglioma). Fifty-two percent (n = 12) of patients were treated following recurrence, while

48% (n = 11) were treated in the newly diagnosed setting. All patients were treated following surgical resection and standard-of-care treatment. The molecular characteristics of the patient tumors are outlined in Table 1. Overall, MGMT methylation was seen in 35% (n = 8), IDH mutations were observed in 35% (n = 8, all grade III), and EGFR amplification was seen in 44% (n = 10, all glioblastoma) of patients, consistent with the heterogenous population of malignant glioma patients. There were no statistically significant differences in age, sex, Karnofsky performance status, MGMT methylation status, pre- or postsurgery enhancing tumor volume, nor steroid administration at enrollment. No statistically significant differences were observed between the molecular characteristics, although the number of patients in each treatment group was small.

Overall, the addition of a TLR agonist-induced only Grade 1-2 treatment-related adverse events (TRAEs), and all adverse events reported resolved without further treatment or hospitalization (Table 2). The most common TRAEs were rash (39%), fever (35%), and fatigue (26%; see Table 2), and were more common in patients treated with resiquimod and poly-ICLC. 88.9% of patients who received resiquimod reported a temporary localized, cutaneous rash that resolved without further treatment. Other observed adverse events were not uncommon in the setting of postoperative central nervous system (CNS) tumor treatment. However, no serious adverse events (Grade 3-4) attributable to the treatment were observed. As such, the addition of a TLR agonist to ATL-DC vaccination in malignant glioma patients was found to be safe and tolerable.

### Adjuvant TLR agonist treatment induces systemic expression of type I and type II interferon downstream genes

The primary endpoint of this clinical trial was to evaluate systemic immune response changes induced by ATL-DC vaccination with and without TLR agonist administration. As such, we collected PBMCs at baseline (pre-treatment), one day after the vaccination (on treatment), and then following the completion of the treatment cycle (post-treatment) of each patient (Fig. 1A). We aimed to understand how the adjuvant administration of TLR agonists modified the immune response in comparison with ATL-DC vaccination alone (placebo control).

We first performed paired bulk RNA-seq on patient-matched, pre-treatment and post-treatment PBMC samples that passed QC (see sample list in Supplementary Data 1C). For each gene, we computed the difference between its expression in the pre- and post-samples of patients in each treatment group: ATL-DC+placebo (n = 5 pairs); ATL-DC+poly-ICLC (n = 8 pairs); ATL-DC+resiquimod (n = 8 pairs); for brevity, we refer to them as placebo, poly-ICLC and resiquimod, respectively. To identify expression changes specific to the TLR agonist groups, we identified genes whose average upregulation in the TLR agonist pairs (poly-ICLC or resiquimod) were at least two-fold higher than the placebo pairs (Fig. 1B, Supplementary Data 2A, see Methods).

Genes upregulated in the TLR agonist groups were involved in antigen processing and were enriched with known interferon-stimulated genes (ISGs) (Fig. 1C–E, Supplementary Data 2B, C). This observation was also confirmed by per-sample gene set enrichment analysis, where the TLR agonist-treated groups displayed higher enrichment of both type I and II interferon downstream gene sets compared to ATL-DC/placebo (Fig. 1F, Supplementary Data 2D, E). PBMC samples with higher absolute enrichment scores of interferon gene sets were dominated by post-treatment samples from both grade III and IV glioma patients in the TLR-agonist-treated groups (Fig. 1G). The two TLR agonist-treated groups showed a largely similar trend in treatment-induced gene expression changes, which included a measurable increase in the expression of ISGs in the peripheral blood of malignant glioma patients. However, we noted that the resiquimod group had a more heterogenous response, which resulted in a lower degree of statistical significance compared to that of the poly-ICLC group.

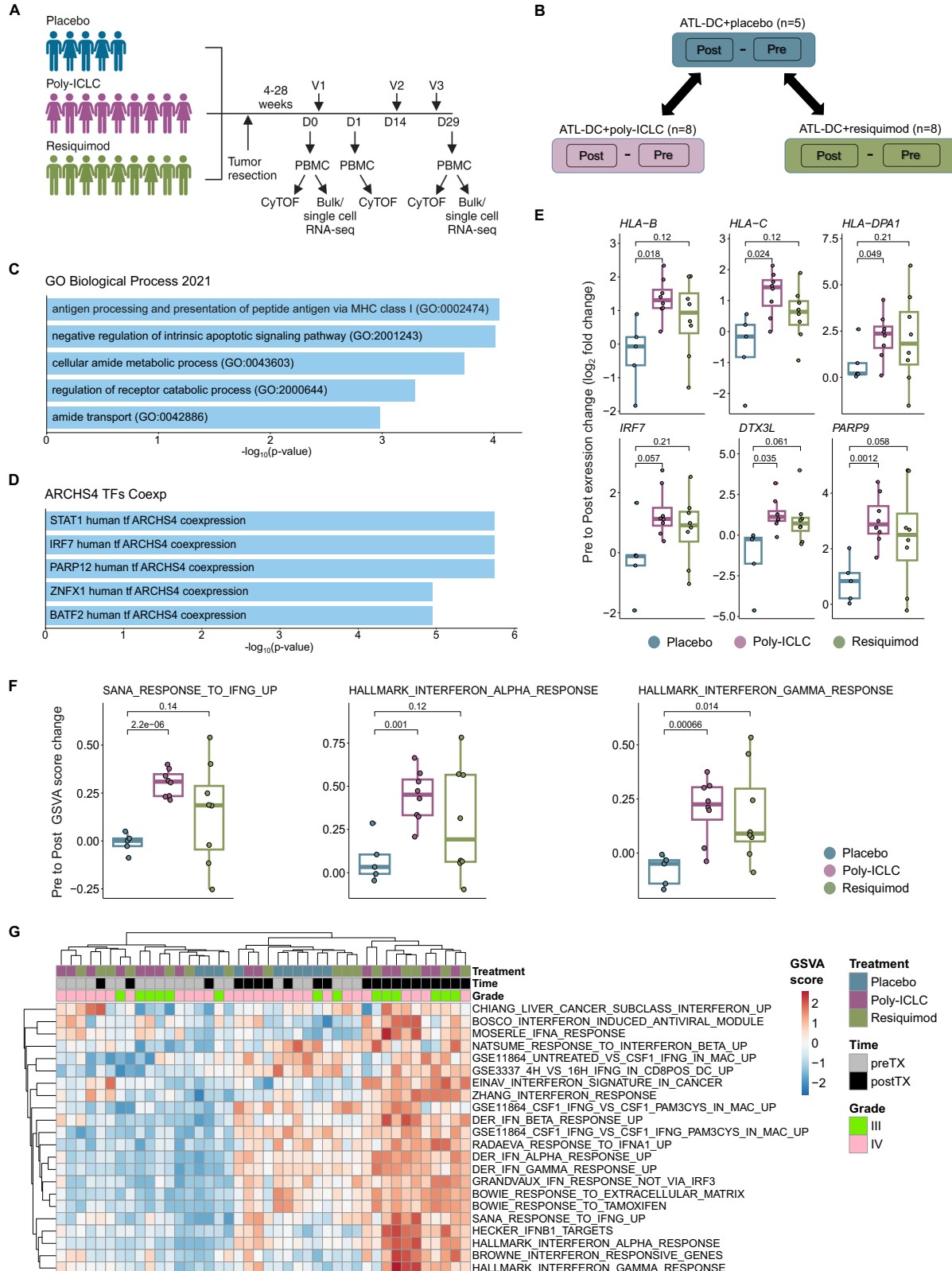

## TLR agonist treatment induces systemic T cell activation, monocyte proliferation and interferon responses in myeloid and lymphoid populations

We performed CyTOF on PBMC timepoints with a 27-marker heavy metal antibody-conjugated panel for 20 of the 23 patients where sufficient material was available (placebo, $n = 4$ pairs; poly-ICLC, $n = 9$ pairs; resiquimod, $n = 7$ pairs; see Supplementary Data 1C, 3A, 3B). The panel was selected to be able to broadly characterize different immune cell types, activation/effector, memory, and exhaustion phenotypes, with a bias towards T-cell relevant markers. The different immune cell type populations were visualized by the uniform manifold approximation and projection (UMAP) method (Fig. 2A), which we broadly assigned to seven different major immune populations based off the normalized heatmap marker expression (Fig. 2B).

**Fig. 1 | Combination of ATL-DC vaccine and TLR agonists results in a robust interferon pathway activation in the patient PBMCs. A** Timeline of PBMC acquisition and analysis using CyTOF and/or RNAseq. V = vaccine, D = Day. (Figure created with the help of BioRender). **B** Schematic of differential gene expression analysis performed on pre-treatment and post-treatment PBMCs of indicated treatment groups. Differentially expressed genes (DEGs) in TLR agonist-treated groups are compared against their changes in the placebo group to identify DEGs specific to the TLR-agonist groups. **C, D** Enriched gene set terms in Gene Ontology Biological Process (**C**) or ARCHS4 TF Coexp (**D**) datasets that significantly overlap with the union of DEGs from ATL-DC + poly-ICLC and ATL-DC + resiquimod groups (*P* values, FDR-adjusted, two-sided fisher exact test). **E** Differential gene expression (pre vs. post-treatment fold change, in log$_2$) of representative antigen presentation and IFN-related genes across treatment groups (P values, two-sided Welch t test).

**F** Gene set enrichment score differences (pre vs. post-treatment, delta GSVA score) of representative IFN-related genesets across treatment groups (*P* values, two-sided Welch t test). **G** Heatmap of single-sample, gene set enrichment scores (GSVA) of type I and type II interferon genesets in pre-treatment, ATL-DC + placebo, ATL-DC +poly-ICLC and ATL-DC+resiquimod samples. The number of sample pairs analyzed in panels **E** and **F** are: ATL-DC+placebo, 5 pairs; ATL-DC+poly-ICLC, 8 pairs; ATL-DC+resiquimod, 8 pairs. The rectangular box in each boxplot represents the interquartile range (IQR), spanning from the first quartile (25$^{th}$ percentile, bottom of box) to the third quartile (75th percentile top of box). Inside the box, the median (50th percentile) is marked. The whiskers (shown as lines extending from the box) extend to the largest and smallest non-outlier values within 1.5 times the IQR, while outliers lie beyond the whiskers.

## Table 1 | Baseline patient characteristics

| Variable | DC vaccine + placebo (*n* = 5) | DC vaccine + poly-ICLC (*n* = 9) | DC vaccine + resiquimod (*n* = 9) | Total (*n* = 23) |
|---|---|---|---|---|
| Age (year) | | | | |
| Mean (SD) | 56.50 (13.75) | 44.09 (12.04) | 43.73 (8.80) | 46.65 (11.98) |
| Median (IQR) | 48.06 (45.7-71.45) | 40.15 (35.35-54.6) | 43.46 (35.55-51.55) | 45.33 (37.4-72.8) |
| Sex | | | | |
| Female, n (%) | 2 (40%) | 5 (56%) | 3 (33%) | 10 (43%) |
| Male, n (%) | 3 (60%) | 4 (44%) | 6 (67%) | 13 (57%) |
| OS (months) | | | | |
| Median (IQR) | 7.7 (7.33-NA) | 52.5 (26.6-NA) | 16.7 (15.33-NA) | 24.5 (15.3-61.2) |
| TTP (months) | | | | |
| Median (IQR) | 5.13 (4.5-NA) | 31.43 (13.4-NA) | 8.1 (6.2-NA) | 8.1 (6.13-31.3) |
| WHO grade, n (%) | | | | |
| III | 1 (20%) | 4 (44%) | 3 (33%) | 7 (30%) |
| IV | 4 (80%) | 5 (56%) | 6 (67%) | 16 (70%) |
| Recurrence, n (%) | | | | |
| None | 1 (20%) | 5 (56%) | 3 (33%) | 9 (39%) |
| Recurrence | 4 (80%) | 4 (44%) | 6 (67%) | 14 (61%) |
| MGMT status, n (%) | | | | |
| Methylated | 1 (20%) | 4 (44%) | 3 (33%) | 8 (35%) |
| Unmethylated | 4 (80%) | 5 (56%) | 6 (67%) | 15 (65%) |
| EGFR classification, n (%) | | | | |
| Amplified | 3 (60%) | 2 (22%) | 5 (56%) | 10 (44%) |
| Not amplified | 1 (20%) | 5 (56%) | 3 (33%) | 9 (39%) |
| Unknown | 1 (20%) | 2 (22%) | 1 (11%) | 4 (17%) |
| IDH status, n (%) | | | | |
| Mutant | 1 (20%) | 4 (44%) | 3 (33%) | 8 (35%) |
| Wild-type | 4 (80%) | 5 (56%) | 6 (67%) | 15 (65%) |
| Avastin treatment, n (%) | | | | |
| Treated | 4 (80%) | 5 (56%) | 6 (67%) | 15 (65%) |
| Not treated | 1 (20%) | 4 (44%) | 3 (33%) | 8 (35%) |

## Table 2 | Adverse events across treatment cohorts

| Variable | DC vaccine + placebo (*n* = 5) | DC vaccine + poly-ICLC (*n* = 9) | DC vaccine + resiquimod (*n* = 9) | Total (*n* = 23) |
|---|---|---|---|---|
| Any | 1 (20%) | 9 (100%) | 8 (89%) | 18 (78%) |
| Rash | 0 | 1 (11%) | 8 (89%) | 9 (39%) |
| Fever | 0 | 5 (56%) | 3 (33%) | 8 (35%) |
| Fatigue | 1 (20%) | 2 (22%) | 3 (33%) | 6 (26%) |
| Flu-like symptoms | 0 | 2 (22%) | 0 | 2 (9%) |
| Nasal congestion | 0 | 1 (11%) | 0 | 1 (4%) |
| Nervous system | 0 | 4 (44%) | 2 (22%) | 4 (17%) |
| Headache | 0 | 3 (33%) | 1 (11%) | 4 (17%) |
| Seizure | 0 | 0 | 1 (11%) | 1 (4%) |
| Sensory paresthesias | 0 | 1 (11%) | 0 | 1 (4%) |
| Cognitive disturbances | 0 | 1 (11%) | 0 | 1 (4%) |
| Ear pain | 0 | 1 (11%) | 0 | 1 (4%) |
| Musculoskeletal | 0 | 3 (33%) | 2 (22%) | 5 (22%) |
| Neck pain | 0 | 0 | 1 (11%) | 1 (4%) |
| Body aches | 0 | 1 (11%) | 0 | 1 (4%) |
| Myalgia | 0 | 2 (22%) | 1 (11%) | 3 (13%) |
| Gastrointestinal | 0 | 1 (11%) | 1 (11%) | 2 (9%) |
| Nausea | 0 | 1 (11%) | 1 (11%) | 2 (9%) |
| Vomiting | 0 | 0 | 1 (11%) | 1 (4%) |
| Cardiovascular / blood | 0 | 0 | 2 (22%) | 2 (9%) |
| Presyncope | 0 | 0 | 1 (11%) | 1 (4%) |
| Neutropenia | 0 | 0 | 1 (11%) | 1 (4%) |

Supplementary Data 3D). ATL-DC + TLR agonist treatment induced PD-1 expression in CD4 T cell population and increased the T-cell normalized expression of *PDCD1* (the transcript that encodes PD-1 protein) and *TCF7* (a marker of progenitor-like T cells) (Fig. 2D, Supplementary Fig. 2C). Moreover, expression of markers associated with irreversible T cell exhaustion, such as CD38 and CD39[33,34], were also significantly reduced after ATL-DC + TLR agonist treatment (Fig. 2D, Supplementary Fig. 2D). Increased expression of PD-1 and decreased expression of CD38 and CD39 suggest the addition of the TLR agonists led to enhanced systemic T cell activity and cellular fitness in the patient.

To delineate the changes induced by ATL-DC and TLR agonist treatment in discrete peripheral blood immune cell subsets, we performed single-cell RNA-seq on selected patients at baseline and then following the completion of therapy. We analyzed two representative sample pairs from each cohort (placebo, poly-ICLC, and resiquimod)

After 3 cycles of treatment, the post-treatment samples of patients in the TLR agonist groups showed a significant increase in the proportion of proliferating Ki67 + CD14+ classical monocytes (Fig. 2C, Supplementary Data 3C). Such findings were corroborated by the increased monocyte fraction and *CD14* transcript expression after ATL-DC + TLR agonist-treated samples (Supplementary Fig. 2A, B,

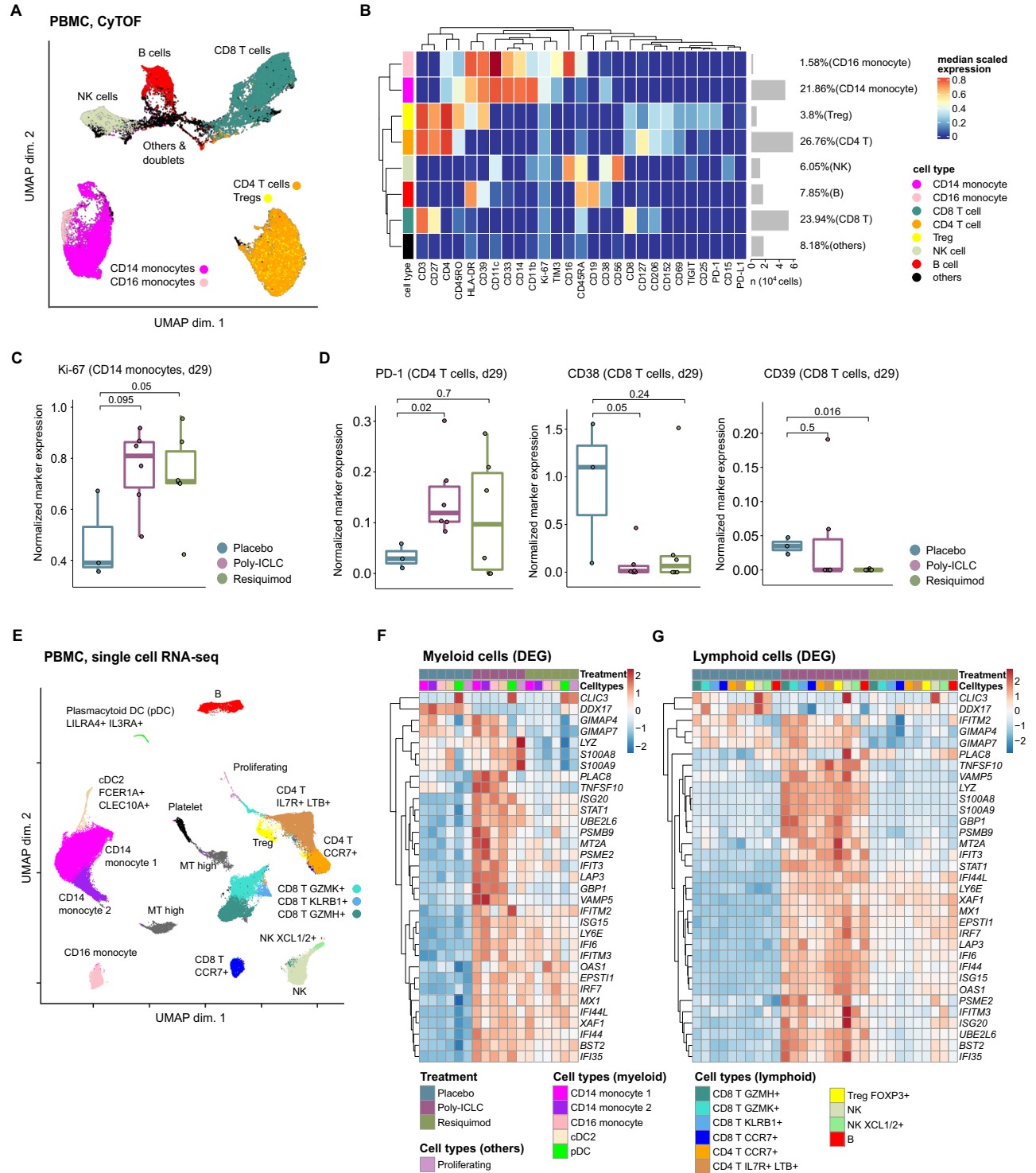

**Fig. 2 | Single cell analysis reveals activation of systemic T cells and monocytes as a part of interferon pathway activation in all myeloid and lymphoid populations. A** A UMAP projection of the pre- and post-treatment PBMC sample pairs from twenty patients (placebo, *n* = 4 pairs; poly-ICLC, *n* = 9 pairs; resiquimod, *n* = 7 pairs). Clustering was performed with a random sampling of 5,000 cells from each patient. **B** Heatmap of normalized expression of all 27 cell markers within cell populations identified in the patient PBMCs. **C, D** Normalized expression of indicated markers in monocyte (**C**), or T cell populations (**D**) within the PBMC samples of patients from indicated treatment groups. P values, two-sided Wilcoxon rank sum test. **E**, UMAP projection of the PBMC-derived single cells (*n* = 99,590). The immune subset associated with each cluster is inferred based on the cluster's differentially expressed transcripts. Canonical markers of known immune subsets are

shown. **F, G** Heatmaps showing the union of recurrent DEGs computed between ATL-DC treated samples (combined with placebo, resiquimod or poly-ICLC) and pre-treatment samples in the myeloid populations (**F**) or lymphocyte populations (**G**). Shown in the heatmaps are the log fold change values of the DEGs in each cell population grouped by their treatment groups. The number of sample pairs analyzed in **C** and **D** are: ATL-DC+placebo, 4 pairs; ATL-DC+poly-ICLC, 9 pairs; ATL-DC+resiquimod, 7 pairs. The rectangular box in each boxplot represents the inter-quartile range (IQR), spanning from the first quartile (25th percentile, bottom of box) to the third quartile (75th percentile the top of box). Inside the box, the median (50th percentile) is marked. The whiskers (shown as lines extending from the box) extend to the largest and smallest non-outlier values within 1.5 times the IQR, while outliers lie beyond the whiskers.

(Supplementary Data 1C, 3E). We identified a total of twelve clusters from the total PBMC immune cell population and annotated these clusters based on differentially expressed gene markers in each cluster. From the initial clustering, we were able to identify multiple populations of CD4$^+$ and CD8$^+$ T cells, two populations of NK cells, three monocytic cell populations, B cell, and dendritic cells (type 2 conventional dendritic cells (cDC2) and plasmacytoid dendritic cells (pDCs), in accordance with the previous characterization of these cell types in peripheral blood (Fig. 2E and Supplementary Fig. 2E, F).

Differential gene expression analysis across the different lymphoid and myeloid populations revealed concordant upregulation of known ISGs (e.g. *IFI6/35/44 L, ISG15/20, IFIT3, IFITM1/3, GBP1/5, MX1, STAT1, and CXCL10*) and antigen presentation-related proteasomes (*PSMB9* and *PSME2*) in both TLR agonist sample pairs. The magnitude of induction was weaker in the paired PBMC samples obtained from the resiquimod group compared to the poly-ICLC group (Fig. 2F, G).

Thus, our combination of high dimensional proteomics, bulk and single-cell RNAseq demonstrates how adjuvant TLR administration in conjunction with ATL-DC reproducibly increases the proportion of canonical CD14+ monocytes within the systemic blood circulation. This TLR agonist administration was also associated with enhanced T cell activity, coupled with decreased expression of CD38 and CD39 and their downstream T cell-suppressive adenosine pathway[33-35]. ATL-DC + TLR agonist-driven induction of ISGs across lymphoid and myeloid populations identified in our scRNAseq analysis corroborated our bulk transcriptomic analysis. Given the consistent changes observed with TLR agonist administration, we examined whether these systemic measurements correlated the observed progression-free and overall-survival differences between these patient populations to speculate on their contribution.

### Long-term clinical outcomes of malignant glioma patients treated with ATL-DC vaccination plus adjuvant TLR agonists

Median follow-up of patients treated on this clinical trial was 2.2 years after surgery, although the long-term survivors have now been followed for over 10 years. Median progression-free survival (PFS) was 8.1 months; and median overall survival (OS) was 26.6 months. Although this clinical trial was not designed or powered to detect effects of these treatments on survival between the treatment groups, there were noticeable differences in median survival between the treatments groups for both OS (placebo: 7.7 months, poly-ICLC: 52.5 months, and resiquimod: 16.7 months; log-rank $P = 0.017$) and PFS (placebo: 5.5 months, poly-ICLC: 31.4 months and resiquimod: 8.1 months; log-rank $P = 0.0012$) (Fig. 3A). Because the trial included patients with both grade III and IV tumors, we stratified our analysis based on tumor grade. When we analyzed only the grade IV (GBM) patients, we observed a trend towards improved PFS (log-rank $P = 0.068$) and OS (P not significant) (Fig. 3B). Interestingly, for the IDH mutant/Grade III cohort, all four patients that received ATL-DC + poly-ICLC treatment are still alive at the data cutoff date (three of the patients have survival > 120 months and one > 112 months), and they have significantly longer OS and PFS compared to the other ($n = 4$) grade III patients who received ATL-DC + resiquimod or ATL-DC alone where median OS was 15.73 months (Fig. 3C).

We performed multivariate Cox proportional hazard (PH) analysis, adjusting for clinical variables that are significantly correlated with OS or PFS as a single variable (tumor grade, MGMT methylation status, and number of recurrences). Our analysis confirmed that patients in the poly-ICLC and resiquimod treatment groups had a lower risk of progression that was independent of grade, MGMT methylation, and number of recurrences (Fig. 3D). Risk of death was significantly lower in the poly-ICLC group, while the resiquimod group showed a similar trend that was not statistically significant (Supplementary Fig. 3A). In the GBM patient subset, TLR agonist treatment also significantly

lowered risk of recurrence, but not risk of death (Fig. 3E, Supplementary Fig. 3B).

To determine whether this treatment directly impacted tumor volume, MR imaging was performed, and contrast-enhancing tumor volume was quantified over time. We noted that the rate of tumor volume increase over time in the ATL-DC/placebo treatment cohort was higher than in the ATL-DC/resiquimod treatment ($p = 0.022$) and the ATL-DC/poly-ICLC treatment groups ($P < 0.001$; Fig. 3F). Anecdotally, we observed an increased T2/FLAIR MRI signal after completion of the vaccine series in two of the four long-term survivors who received ATL-DC/poly-ICLC (Supplementary Fig. 3C, D), although such findings are potentially confounded by prior radiation therapy, and thus we cannot ascribe such changes solely to the vaccine/TLR agonist intervention. However, this increased post-vaccination T2/FLAIR on MRI was not seen in patients who did not receive poly-ICLC (not shown).

### Interferon activation score in the peripheral blood immune cells is a significant predictor of survival after ATL-DC therapy

Finally, we asked if the magnitude of interferon pathway induction by the adjuvant TLR agonist treatment directly correlated with OS or PFS. This could allow for the use of an interferon activity score as a biomarker for productive anti-tumor immune responses following ATL-DC immunotherapy. To this end, we stratified the patients by the median GSVA score of the "HALLMARK INTERFERON GAMMA RESPONSE" gene set in post-treatment PBMC samples. We confirmed that patients whose post-treatment samples displayed higher interferon gene set scores (≥median) had longer OS and PFS than those with lower scores (Fig. 4A, Supplementary Fig. 4A). Separate analyses on the grade IV (GBM) and grade III glioma patients showed a concordant trend but with a lower degree of statistical significance; this was likely caused by the small sample sizes. Notably, multivariate Cox PH analysis strongly suggested that the interferon gene set score is a significant predictor of tumor recurrence (Fig. 4B, C) and death (Supplementary Fig. 4B), even after adjusting for other potentially confounding clinical variables. To ensure that the correlation is not specific to this single gene set, we confirmed that the gene set scores of other interferon gene sets after ATL-DC treatment are also positively correlated with the patients OS and PFS (Supplementary Data 4A, B). Such findings can be confirmed in larger subsequent studies.

Taken together, these data suggest that the addition of TLR agonists to ATL-DC vaccination shifts towards an interferon-induced immune response in both lymphoid and myeloid cells. Poly-ICLC and resiquimod appear to upregulate similar ISGs but with different magnitude. Enhancing systemic ISG-signaling may reflect an environment more favorable towards the generation of an antitumor immune response and clinical effects.

## Discussion

We report herein that ATL-DC vaccination with adjuvant poly-ICLC or resiquimod is overall safe and well-tolerated in patients with malignant glioma. To achieve the primary immunological endpoints of this study, we utilized high-dimensional single-cell analysis to understand the systemic proteomic and transcriptomic changes induced by TLR agonists in order to rationally determine the optimal therapeutic combination.

Our study is the first high-dimensional single-cell analysis done in a clinical trial for malignant glioma patients treated with dendritic cell vaccination and TLR agonists. Although our study was not designed to examine what happens in the tumor microenvironment, our results indicate that we are able to sensitively detect systemic changes in the blood after intradermal autologous dendritic cell vaccination with and without TLR agonists. Adjuvant TLR agonist treatment promotes the expression of IFNα/β and IFNγ-induced genes on the peripheral lymphoid and myeloid cells, and GSVA further confirmed increased

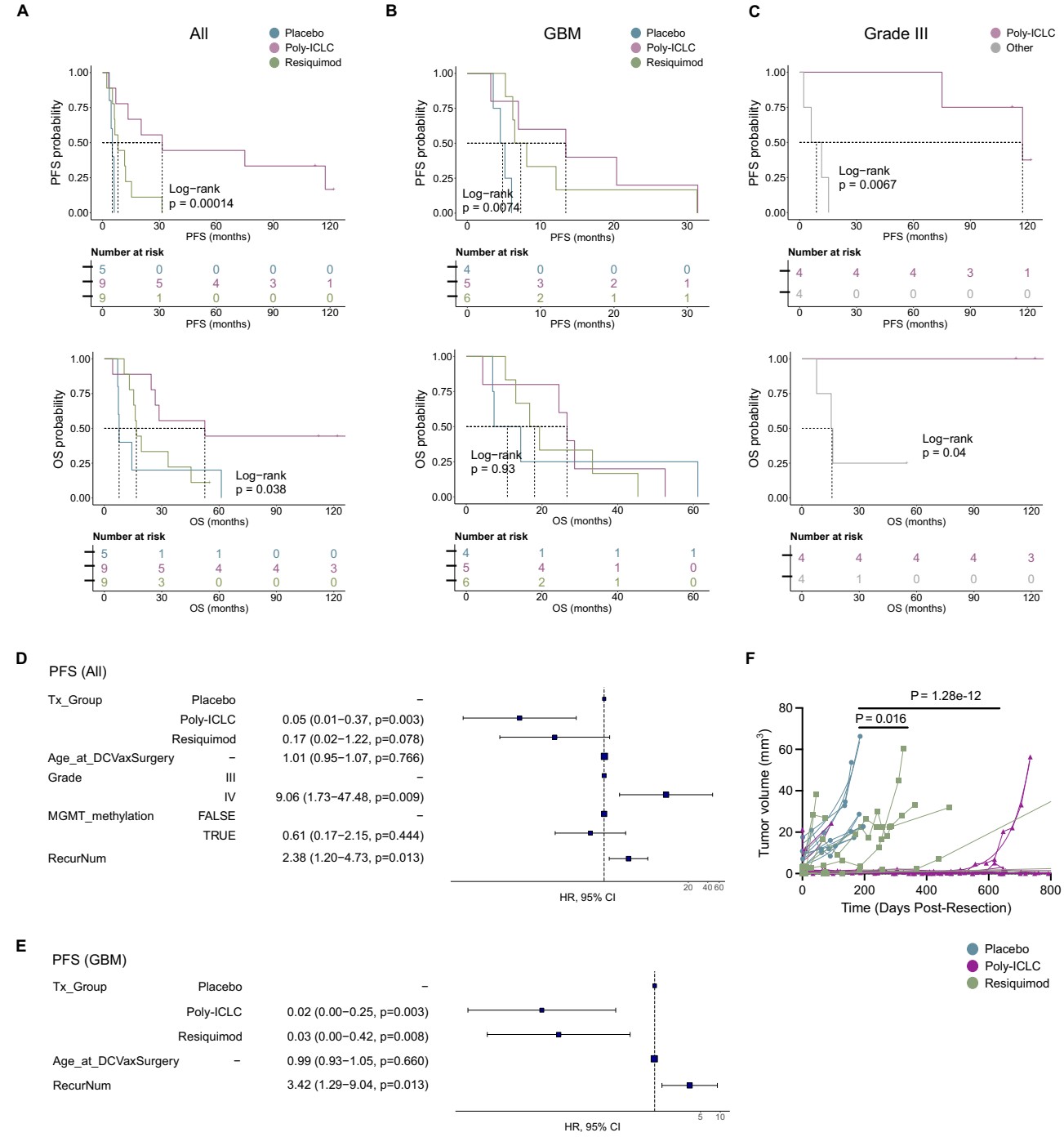

**Fig. 3 | Combined ATL-DC vaccine and TLR agonist treatment show trends of improved tumor control and patient survival. A–C** Progression-free survival (PFS, top) and overall survival (OS, bottom) of all patients (**A**), patient subset with GBM (**B**), or grade III glioma (**C**) in indicated treatment groups. *P* values, log-rank test. **D, E**, Multivariate Cox proportional hazards analysis assessing the hazard ratios of tumor progression in TLR agonist treatment groups against placebo in all patients (**D**) or GBM subset (**E**) after adjusting for other clinical covariates (Tx_Group=treatment group, RecurNum=number of recurrences prior to ATL-DC treatment). In the forest plot, the squares are the hazard ratio (HR) estimates, the error bars are 95% confidence interval (CI) of the HR, the *P* value of each covariate is based on its Wald statistics, the *P* values are not adjusted. In **D**, the sample distribution in each covariate is Tx_Group: placebo=5, poly-ICLC = 9, resiquimod=9; Grade: III = 8, IV = 15; MGMT_methylation: True=8, False=15. In **E**, Tx_Group: placebo=4, poly-ICLC = 5, resiquimod=6. **F**, MR-computed volumes of post-treatment, recurrent tumors in indicated treatment groups. Treatment groups: Placebo (*n* = 5), Resiquimod (*n* = 8); Poly ICLC (*n* = 9). *P* values, unpaired, two-sided Wilcoxon rank sum test.

expression of the IFNα and IFNγ downstream genes, including IFN-induced proteins *ISG15* and *STAT1*. Other genes that were significantly upregulated by TLR agonist treatment include PARP9-DTX3L, and this heterodimer is also known to amplify interferon signalling[36]. Our current observation of increased mRNA expression of interferon downstream genes may represent activation of either type I or type II interferons or both; type I and II interferon downstream gene signatures significantly overlap with one another, and additional follow-up study is required to dissect the individual contribution of type I and type II interferons in the efficacy of ATL-DC therapy. Regardless, our

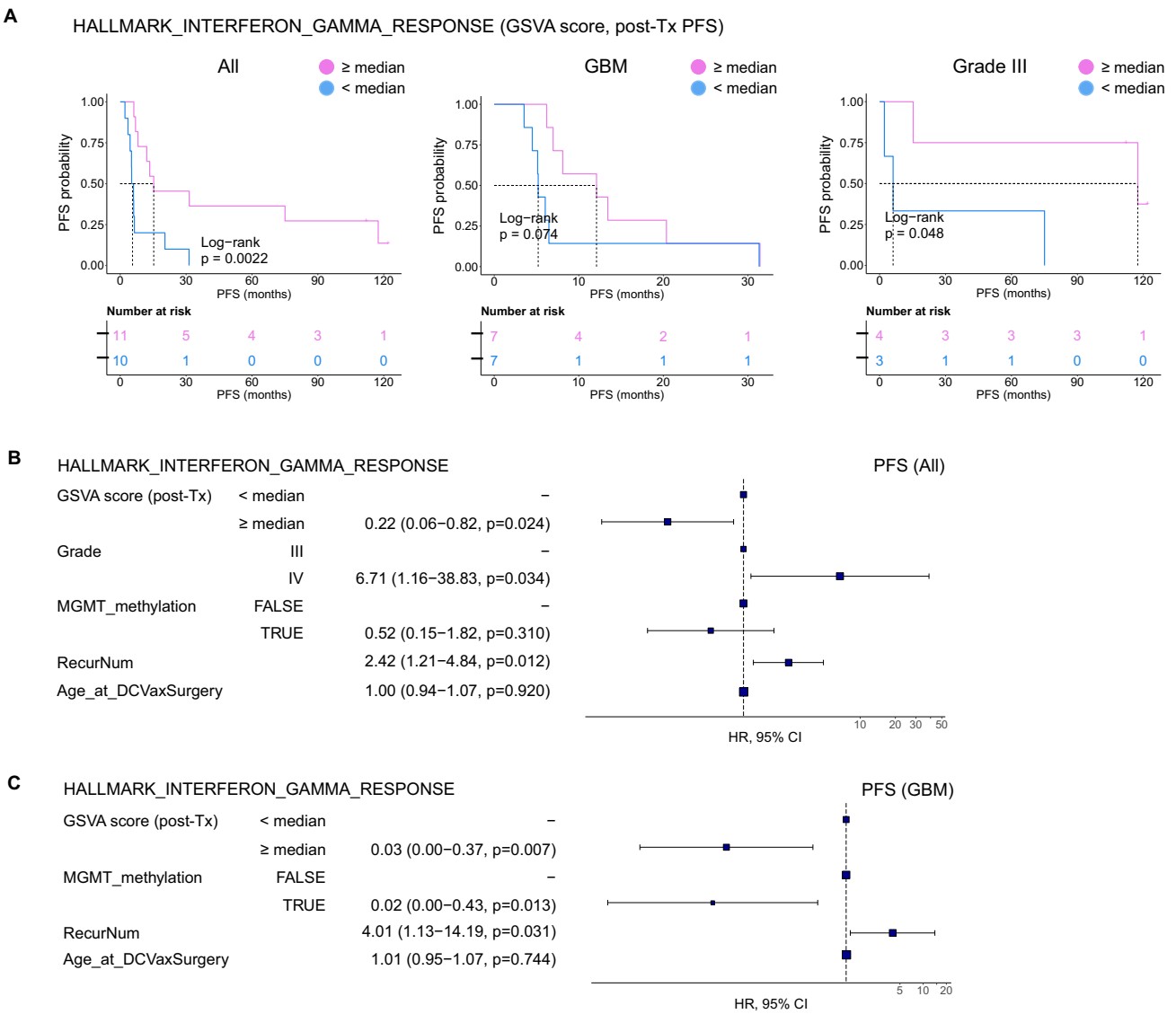

**Fig. 4 | IFN pathway activation is a positive predictor of survival after ATL-DC vaccine and TLR agonist therapy. A** Kaplan-Meier progression-free survival curves of all patients (left), GBM (center), and Grade III glioma subsets (right) stratified by their HALLMARK_INTERFERON_GAMMA_RESPONSE GSVA scores in their post-treatment PBMCs. *P* values, log-rank test. **B, C** Multivariate Cox proportional hazards analysis assessing hazard ratios of tumor progression in patients with high HALLMARK_INTERFERON_GAMMA_RESPONSE GSVA score in all patients (**B**) or GBM subset (**C**) after adjusting for other clinical covariates. In the forest plot, the squares are the hazard ratio (HR) estimates, the error bars are 95% confidence interval (CI) of the HR, the *P* value of each covariate is based on its Wald statistics, the *P* values are not adjusted. In **B**, the sample distribution in each covariate is GSVA score (post-Tx): <median=10, ≥median=11; Grade: III = 7, IV = 14; MGMT_methylation: True=7, False=14. In (**C**), GSVA score (post-Tx): <median=7, ≥median=7; MGMT_methylation: True=3, False=11.

results support the conclusion that DC vaccination with poly-ICLC induces Type I and Type II IFN responses more effectively than with adjuvant resiquimod or a dendritic cell vaccine alone. Similar to our results, additional studies have identified poly-ICLC as the most effective TLR/PRR agonist when compared with others[37,38]. The downstream effect of this signaling in the lymphoid cell population appears to be increased T-cell activity, as well as decreased T-cell exhaustion phenotype. Together, these effects may enhance the activity of tumor antigen-specific T cells generated from an active vaccine.

It is also important to recognize that, in contrast to resiquimod and even plain poly-IC, poly-ICLC signals through various PRRs in addition to TLR3, consistent with its role as an authentic viral mimic. The poly-lysine stabilizer also functions as a transfection agent. Specifically, poly-lysine bursts the endosome through a proton sponge effect and releases the dsRNA into the cytoplasm, where it then

preferentially activates MDA5, OAS, PKR, and other cytoplasmic dsRNA-dependent systems[21]. Among the actions generated putatively through MDA5 are a further increase in Type 1 IFN, depression of MDSC, expansion of CD8 T cell populations through IL-15, CD8 targeting and infiltration of tumor through CXCL10, and a direct Type 1 IFN-dependent effect on tumor endothelium through VCAM-1[39]. These effects are best seen with systemic (intramuscular or intravenous) rather than local (subcutaneous) administration, as we have done in the current study. Such adjuvant responses induced by poly-ICLC may play a role in the longer-term maintenance of the immune responses generated by ATL-DC vaccination, but further studies are required to verify these findings.

While malignant gliomas are usually conceived of as a locor-egional disease with essentially no capacity to spread outside the central nervous system, there has been a growing understanding of the role that systemic tissues play in priming, developing and/or

suppressing an immune response in the brain. The catalog of known pervasive systemic immune deficits in glioblastoma patients is continually growing[40]. The failure of immune checkpoint inhibitor therapy in malignant gliomas has led many to conclude that immune cells in the tumor microenvironment of cancers unresponsive to these checkpoint inhibitors may exist in an irreversible, terminally exhausted state[41,42]. The generation of de novo tumor antigen-specific immune responses in the periphery that lead to new T-cell infiltration into the tumor microenvironment may be required to overcome this barrier[43]. Dendritic cell vaccines are a robust example of an agent capable of mediating the initiation of such a T-cell response.

The fraction of monocytes in the systemic circulation is known to be an important biomarker for the response to PD-1 checkpoint blockade immunotherapy[44]. Our data is consistent with other findings that TLR agonists can induce a higher fraction of CD14+ classical monocytes in the blood. These findings further suggest that the combination of ATL-DC + TLR agonist with immune checkpoint blockade may be a rational choice. In fact, we have now initiated a phase I trial combining ATL-DC+Poly-ICLC with pembrolizumab in recurrent glioblastoma patients (NCT04201873). Our data also reveals increased relative expression of *PDCD1* and *TCF7* after ATL-DC/poly-ICLC. The abundance of TCF7 + PD1 + CD8 T cells was associated with better response to PD-1 blockade in melanoma and non-small cell lung cancer[45,46].

While encouraging, our clinical findings must be interpreted with caution. Even though this was a randomized clinical trial (randomization software assigned patients to TLR agonist/placebo groups), the small number of patients enrolled contributed to an imbalance in patient selection between the treatment groups. Such effects are inherent in trials with small numbers of patients. The patients randomized to the resiquimod group and poly ICLC group were approximately consistent, but the patients in the ATL-DC + placebo had more unfavorable clinical characteristics; the distribution of patient age and disease grade in the placebo group are more unfavorable. We found that ATL-DC vaccinated patients randomized to receive adjuvant TLR agonists demonstrated a statistically significant extended overall time to tumor progression and slower rates of tumor growth, compared with those who received an adjuvant placebo. The poly-ICLC group was further associated with a statistically significant increase in overall median survival. Some of the grade III gliomas treated with ATL-DC/poly-ICLC exhibited unique T2/FLAIR changes on brain MRI scans following DC vaccination, but such findings were confounded by previous radiation therapy, even though such changes were not seen in the other patients. The significance of these imaging findings is not clear and needs to be replicated.

In conclusion, we demonstrate that autologous dendritic cell vaccination plus TLR agonists in patients with malignant gliomas generates a systemic interferon activation signature in the peripheral blood that is correlated with overall survival. Although this was a randomized study, it was powered for immune biomarker analysis, not for survival. As such, the clinical efficacy outcomes should be interpreted with caution. Given the noted long-term survival with the adjuvant use of poly-ICLC with DC vaccination, particularly in the grade III cohort of patients, further clinical trials that incorporate these combinations of immunotherapeutic agents are warranted.

## Methods
### Study design
This was a single-center, randomized, open-label multi-arm phase II clinical trial. The study protocol was approved by an independent ethics committee, institutional review board and internal scientific peer review committee at the University of California, Los Angeles. Patients were recruited and completed treatment between 2010 and 2014, with survival follow-up until the present date. Subjects were not compensated, and all patients gave written informed consent before enrollment.

Twenty-three patients with high-grade WHO Grade III or IV gliomas were enrolled in this protocol. To be eligible for the primary cohort, patients had to be >18 years and have newly diagnosed or recurrent WHO Grade III or IV malignant glioma, as determined through central pathology review. For all patients, a Karnofsky Performance Score (KPS) of ≥60, adequate bone marrow, liver, and renal function, life expectancy of ≥8 weeks, no other prior malignancy within the last 5 years, no active viral infections, and sufficient resected tumor material to produce the autologous vaccine were required. All newly diagnosed patients underwent surgical resection followed by radiation and chemotherapy with temozolomide for 6 weeks, per standard of care. Patients in the recurrent setting proceeded to trial treatment after recovery from surgery. All patients were scheduled to receive ATL-DC. Patients were then randomized to receive either placebo, resiquimod (topical 0.2%, 3 M), or poly-ICLC (20 µg/kg i.m., Oncovir) as an adjuvant to the DC vaccine. Patients underwent leukapheresis to obtain adequate numbers of PBMC for DC generation. For the study treatment, we processed the resected tumor tissue into a lysate, then prepared and cryopreserved the autologous DCs as we previously described[2,3]. Patients were then treated with three intradermal injections of autologous tumor lysate-pulsed DC plus adjuvant TLRs/placebo on days 0, 14, and 28. The TLR agonists were delivered as separate injections. Poly ICLC (20 ug/kg) was given as an intramuscular injection at the same site as the intradermal ATL-DC vaccine. Resiquimod (0.2%) was applied as a topical gel directly over the intradermal ATL-DC vaccine site. The placebo was a gel without any resiquimod and administered similarly over the intradermal vaccine site. All vaccines were administered on the upper arm. Follow-up for patients was conducted at the study site for vital signs, KPS, hematology and serum chemistries, as well as neurological and physical examinations.

### Clinical assessments
Safety was assessed on the basis of occurrence of adverse events, which were categorized according to the NCI Common Toxicity Criteria for Adverse Events v. 4.0. Safety assessments were performed on the day of vaccination and 1 week after each vaccination during the treatment phase, and every 2 months thereafter until tumor progression or death.

Anatomic MR images were acquired prior to DC + adjuvant treatment and at 2-month intervals for all patients using the standardized brain tumor imaging protocol (BTIP)[47], including three-dimensional pre- and post-contrast T1-weighted images at 1-1.5 mm isotropic resolution, two-dimensional T2-weighted and T2-weighted fluid-attenuated inversion recovery (FLAIR) images with 3-4 mm slice thickness and no interslice gap, and diffusion-weighted images with $b = 0$, 500, and 1000 s/mm², 3–4 mm slice thickness and no interslice gap. Disease progression was determined using the modified RANO criteria[48]. Additionally, post-hoc quantitative tumor volumetric analysis was performed using contrast-enhanced T1-weighted digital subtraction maps and segmentation techniques described previously[49–51]. Briefly, linear registration was first performed between all images, including contrast-enhanced T1-weighted images and T2-weighted and/or FLAIR images to nonenhanced T1-weighted images using a 12-degree-of-freedom transformation and a correlation coefficient cost function. Next, intensity normalization and bias field correction were performed for both nonenhanced and contrast-enhanced T1-weighted images, and voxel-by-voxel subtraction between normalized nonenhanced and contrast-enhanced T1-weighted images was performed. Image voxels with a positive (greater than zero) before-to-after change in normalized contrast enhancement signal intensity (i.e., voxels increasing in MR signal after contrast agent administration) within T2-weighted FLAIR hyperintense regions were isolated to create the final T1 subtraction maps. Estimates of tumor volume included areas of contrast enhancement on T1 subtraction maps, including central necrosis (defined as being enclosed by contiguous, positive-enhancing disease).

## Patient samples

Heparinized peripheral blood was collected at the baseline visit and at each treatment visit for immune monitoring. Peripheral blood mononuclear cells were collected in CPT tubes (BD Biosciences, cat: 362753), isolated according to the manufacturer's protocol, placed in freezing media made of 90% human AB serum (Fisher Scientific, cat. MT35060CI) and 10% dimethyl sulfoxide (Sigma, cat. C6295-50ML) and stored in liquid nitrogen until the time of analysis. On the day of data acquisition, samples were thawed in a 37 °C water bath and washed in RPMI-1640 media (Genesse Scientific, cat: 25-506) supplemented with FBS and penicillin and streptomycin. Patient tumor samples were attained immediately following surgery.

## Generation of autologous dendritic cell vaccines

Monocyte-derived DCs were established from adherent peripheral blood mononuclear cells (PBMC) obtained via leukapheresis performed at the UCLA Hemapheresis Unit, as we have published previously[3,6,52]. All ex vivo DC preparations were performed in the UCLA-Jonsson Cancer Center GMP facility under sterile and monitored conditions. In brief, dendritic cells were prepared by culturing adherent cells from peripheral blood in RPMI-1640 (Gibco) and supplemented with 10% autologous serum, 500 U/mL GM-CSF (Leukine®, Amgen, Thousand Oaks, CA) and 500 U/mL of IL-4 (CellGenix), using techniques described previously[2]. Following culture, DCs were collected by vigorous rinsing and washed with sterile 0.9% NaCl solution. The purity and phenotype of each DC lot was also determined by flow cytometry (FACScan flow cytometer; BD Biosciences, San Jose, CA). Cells were stained with FITC-conjugated CD83, PE-conjugated CD86 and PerCP-conjugated HLA-DR mAb's (BD Biosciences). Release criteria were >70% viable by trypan blue exclusion, and >30% of the large cell gate being CD86+ and HLA-DR+. One day before each vaccination, DC were pulsed (co-cultured) with tumor lysate overnight, washed, and the final product was tested for sterility by Gram stain, mycoplasma, and endotoxin testing prior to injection.

## Molecular and immune analyses

**CyTOF.** Cells for mass cytometry analysis were prepared according to the Maxpar cell surface staining protocol. Briefly, 0.5 to $3 \times 10^6$ cells were washed with PBS and treated with 0.1 mg/mL of DNAse I Solution (StemCell Technologies, cat: 07900) for 15 minutes at room temperature. Cells were then resuspended in 5 μM Cell-ID cisplatin (Fluidigm, cat: 201064) as a live/dead marker for 5 minutes at room temperature. After quenching with the Maxpar cell staining buffer (Fluidigm, cat: 201068), the cells were incubated with a 27-marker panel for 30 minutes at room temperature. After washing, cells were incubated overnight in 125 nM iridium intercalation solution (1000X dilution of 125 μM Cell-ID Intercalator-Ir; Fluidigm, cat: 201192 A) in Maxpar Fix and Perm Buffer (Fluidigm, cat: 201067) to label intracellular DNA. The next morning, cells were washed with cell staining buffer and distilled water. The samples were processed on a Helios mass cytometer (Fluidigm) in the University of California, Los Angeles Jonsson Comprehensive Cancer Center Flow Cytometry core.

The CyTOF data was normalized utilizing EQ four-element calibration beads (Fluidigm, cat: 201078) with the R package *premessa* (version 0.2.4, Parker Institute for Cancer Immunotherapy) following removal of dead cells. A total of 5,000 cells were subsampled from each sample (except for sample S16-07-2-Day 1 where we only had 4,861 cells). Subsequently, bead normalized data from 45 samples were integrated as described previously[53]. Briefly, flow cytometry standard (FCS) files were loaded into R with the *flowCore* package (version 2.8.0).

Raw marker intensities were transformed utilizing hyperbolic inverse sine (arcsinh) with cofactor of 5. Cell population identification was carried out using unsupervised clustering using *FlowSOM* package (version 2.4.0) and subsequent metaclustering using

ConsensusClusterPlus package (version 1.60.0). The metaclusters were manually curated to identify canonical populations in Fig. 2B (including one unknown cluster with little/no marker expression). The high dimensional data was visualized with the Uniform Manifold Approximation and Projection (UMAP). Differential marker analysis across treatment groups were first performed using the linear mixed model analysis pipeline as described[53]. Markers with nominally significant p-values in one or more cell populations ($P \le 0.05$; e.g CD39, CD38, Ki-67, PD-1) were visualized in boxplots; statistical significance computed using the linear mixed model were further confirmed using non-parametric Wilcoxon rank sum test.

**Bulk RNAseq.** Total RNA was isolated from frozen PBMC of the patients isolated at baseline and after three biweekly vaccines with ATL-DC plus adjuvant using the ZYMO quick RNA extraction kit. We utilized the TruSeq RNA exome kit to construct the RNA sequencing libraries in samples that passed QC (placebo: 5 pairs, resiquimod: 8 pairs, and poly-ICLC: 8 pairs; see Supplementary Data 1C). Paired-end, $2 \times 100$ base pair (bp) transcriptome reads were mapped to the Genome Reference Consortium Human Build 38 (GRCh38) reference genome using HISAT2 (version 2.0.6)[54]. The gene-level counts were generated by the HTSeq-count program (version 0.5.4p5)[55]. We utilized the DESeq2 R package's counts function (version 1.24.0)[56] to compute the normalized gene expression values from the raw gene expression counts. DESeq2 normalized gene expression was log2 transformed after adding a pseudo count of 1. For subsequent differentially expressed genes (DEGs) and gene set enrichment analyses, we only included the known genes (i.e., genes with RefSeq transcripts ID starting with "NM_", that satisfy: 1) normalized expression IQR ≥1; and 2) normalized log2 expression ≥1 in at least one of the samples.

Based on the filtered gene list, we first obtained the patient-specific, log2 fold change of each gene before and after the ATL-DC vaccine treatment. Next, the mean of the log2 fold changes in the poly-ICLC or resiquimod group is compared to those in the placebo group. Genes showing at least 2-fold upregulation in any of the TLR agonist-treated group (resiquimod or poly-ICLC, nominal t-test $p$ value ≤ 0.05) with respect to the placebo were tested for significant overlap with gene ontology and known gene sets using the web-based tools, ENRICHR[57].

To calculate single sample gene set enrichment of the interferon-related genes, we used the Gene Set Variation Analysis (GSVA) package (version 1.32.0)[58]. To compute the GSVA scores, the filtered, log2 normalized gene expression were supplied to the GSVA program using the 'kcdf=Gaussian' mode. We manually selected gene sets that are related to interferon pathway activation from the c2.cgp, c6, c7, and hallmark geneset collections of the Broad Institute's Molecular Signatures Database (version 7.0)[59].

**Single-cell RNA-seq sample processing and data analysis.** The cells for scRNAseq analysis were resuspended in PBS at a concentration of 1,000 cells/μl. We only selected representative patients from each treatment group whose PBMC quality were sufficient for single-cell RNAseq processing. Cell preparation, library preparation, and sequencing were carried out according to Chromium product-based manufacturer protocols (10X Genomics), targeting for a total of 10,000 cells sequenced. Single-cell RNA sequencing was carried out on a Novaseq 6000 S2 2 x 50 bp flow cell (Illumina) utilizing the Chromium single cell 3' gene expression library preparation (10X Genomics).

The data was aligned with Cell Ranger (version 3.1.0) and aligned to the Genome Reference Consortium Human Build 38 (GRCh38). Data was imported into R (version 4.2.1) and analyzed with the Seurat package (version 4.2.0)[60]. For quality assurance, cells with greater than 20% mitochondrial features were excluded from further analysis. We analyzed a total of 99,590 cells after the QC step. The Seurat data

object from each sample were then integrated and scaled, regressing out the percent mitochondrial features and cell cycle score difference, as described (https://satijalab.org/seurat/index.html). We manually identified each cluster using the genes that were differentially expressed as determined by *FindAllMarker* function; they are visualized using R's ggplot2 and pheatmap packages. Differentially expressed genes (DEGs) corresponding to each treatment group (Placebo vs. Poly-ICLC vs. resiquimod) were computed by first computing cluster-specific DEGs between each group against the pre-treatment (Day 0) samples. To account for intrapatient correlation among cells from the same patient, we computed the DEGs using the *FindAllMarker* function setting the *use.method* parameter to MAST and the *latent.vars* parameter to the patient IDs. The union of cluster-specific DEGs that were seen in at least 25% of all comparisons (the total number of comparisons is the number of treatment groups (3 groups) times the number of lymphoid or myeloid clusters) were selected as recurrent DEGs shown in the heatmaps of Fig. 2F and G.

### Statistical analysis

For the percentage comparisons in the CyTOF analysis, we used the Wilcoxon rank sum test for non-parametric data for 2 independent samples and compared the baseline (Day 0) to Day 1 or Day 29. We performed Fisher's exact test for testing the null of independence of the phenotypic and genotypic characteristics and treatments using the *stats* package in R. Differences in transcript expression log2 fold changes and GSVA scores in the bulk RNA-seq data were calculated with unpaired T test with nonequal variances (two-sided Welch t test). The differences in overall survival or time to progression following treatment (either combination of ATL-DC and placebo, ATL-DC and adjuvant poly-ICLC or ATL-DC and adjuvant resiquimod treatment) were compared using the log-rank test and graphical evaluation of these curves were assessed using the methods of Kaplan and Meier (survminer R package). We further performed multivariable cox proportional hazard (cox PH) regression analysis with HRs (95% CIs) to determine if any of the treatment regimens were significantly predictive of overall survival or time to progression after adjusting for clinical covariates, such as WHO grade, number of recurrences, and MGMT status. The association between interferon pathway score and overall survival or time to progression was analyzed similarly using log-rank (univariate) and Cox PH (multivariate) analyses.

## Data availability

Bulk and single-cell RNA sequencing data are available at Gene Expression Omnibus under accession ID GSE237581. The CyTOF data is uploaded to flowRepository with accession ID FR-FCM-Z6LY.

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

## Acknowledgements

This study was funded in part by the National Institutes of Health NIH/NCI grant (R01CA123396), the UCLA SPORE in Brain Cancer (P50CA211015), NIH National Center for Advancing Translational Science - UCLA CTSI (UL1TR001881), the Parker Institute for Cancer Immunotherapy, and the Brain Tumor Funder's Collaborative. Mass cytometry was performed in the UCLA Jonsson Comprehensive Cancer Center (JCCC) Flow Cytometry Core Facility that is supported by NIH award P30 CA016042. W.H is supported by the NIH/NCI grant (1R01CA236910), Jonsson Comprehensive Cancer Center and the Parker Institute for Cancer Immunotherapy at UCLA. L.S. was supported by a Career Enhancement Program award from the UCLA SPORE in Brain Cancer. A.L. was supported by the UCLA Tumor Immunology Training Grant (USHHS Ruth L. Kirschstein Institutional National Research Service Award # T32 CA009120). L.D is supported by grant from Parker Institute for Cancer

Immunotherapy at UCLA and a postdoctoral fellowship from National Cancer Center.

## Author contributions

R.M.P., L.M.L, A.S., and T.F.C. designed the clinical study. L.M.L., R.G.E., R.M.P., E.B-Y, collected clinical data. W.H., L.S. designed the overall computational analyses. L.D. and W.H. analyzed the bulk RNA-seq. L.S., A.L., R.G.E. M.B., S.K. and C.C. performed single-cell RNA-seq and CyTOF analyses. B.M.E, J.A., and R.G.E. analyzed the MRI data. R.G.E., W.H., L.M.L., and R.M.P. wrote and edited the manuscript. All authors reviewed and approved the manuscript.

## Competing interests

Andres Salazar is the Founder, CEO and Scientific Director for Oncovir, which provided the TLR agonist (Poly-ICLC) used in the trial. Linda M. Liau is a member of the Scientific Advisory Board for Northwest Bioe-therapeutics, Inc, which has licensed the DC vaccine technology. All others declare no competing interests.

## Additional information

[1]Department of Neurosurgery, David Geffen School of Medicine at UCLA, University of California Los Angeles, Los Angeles, CA 90095, USA. [2]Jonsson Comprehensive Cancer Center, David Geffen School of Medicine at UCLA, University of California Los Angeles, Los Angeles, CA 90095, USA. [3]Department of Medicine, Division of Dermatology, David Geffen School of Medicine at UCLA, University of California Los Angeles, Los Angeles, CA 90095, USA. [4]Parker Institute for Cancer Immunotherapy, David Geffen School of Medicine at UCLA, University of California Los Angeles, Los Angeles, CA 90095, USA. [5]Department of Molecular and Medical Pharmacology, David Geffen School of Medicine at UCLA, University of California Los Angeles, Los Angeles, CA 90095, USA. [6]Oncovir, Inc., Winchester, VA, USA. [7]Department of Radiological Sciences, David Geffen School of Medicine at UCLA, University of California Los Angeles, Los Angeles, CA 90095, USA. [8]Department of Neurology/Neuro-Oncology, David Geffen School of Medicine at UCLA, University of California Los Angeles, Los Angeles, CA 90095, USA. [9]These authors contributed equally: Richard G. Everson, Willy Hugo. ✉e-mail: lliau@mednet.ucla.edu; RPrins@mednet.ucla.edu

