## [Peer Review File · Nature Communications]

REVIEWERS' COMMENTS

Reviewer #1 (Remarks to the Author):

The authors have done an excellent job answering all previous comments

Reviewer #2 (Remarks to the Author):

The authors have addressed all of my concerns as best as possible with the data and information they have. I think the weak conclusions are adequately presented as being weak with caveats sufficiently described/cautious language used in interpretation. I was not able to find any additional issues with the MS.

So in summary, This is worthy of publication in Nat Comm, the concerns have been addressed, and I have no additional issues with the manuscript.

Finally, I would like to thank the authors for their diligent revisions.

Reviewer #3 (Remarks to the Author):

In this resubmission of the manuscript looking at clinical immune responses to different ATL-DC-based regimens in glioma patients, the authors have been very responsive to all of the comments made from each of the reviewers.

In particular, they have effectively softened the interpretation overall with inclusion of important caveats in key places when discussing the results and findings of these analyses. They have also effectively clarified the rationale for reporting the results at this point in the study with the numbers of patients, and that this is a precursor to subsequent studies to further evaluate and confirm these findings. Finally,

it is appreciate the additional modeling that was used to help account for important potential sources of bias such as latent confounders and inpatient variability. They have put in a lot of work and the result is a very nice and improved paper.

While overall very responsive to the comments, there are only a few minor issues that still need to be fixed for the final manuscript:

1. For comment #3d, the authors have generated Kaplan-Meier (KM) estimates for the OS and TTP for the different groups; however, this really should replace the values that are currently included in Table 1 or these OS and TTP estimates should be removed altogether from a table focused on baseline characteristics. Thus, I would recommend simply removing OS and TTP from Table 1 altogether, but if it really is of interest to include this here, then the KM estimates (medians and corresponding 95% confidence intervals) should be used instead in this table. This should be an easy change.

2. While caveats have effectively been included, lines 282-284 should be softened as well. the analyses here are not confirming significant predictors, but show a potentially useful predictor even when adjusting for other factors. This can be confirmed in larger subsequent studies.

3. In the statistical methods lines 575-576, the OS and TTP distributions were *compared* using the log-rank test and graphical evaluation of these curves were assessed using the methods of Kaplan and Meier. The survminer R package is just applying this methodology, and it can be referenced parenthetically that it was used, but the methods were those of Kaplan and Meier. This is again an easy fix and just improves the specific accuracy in describing the methodology.

Final Reviewers' Comments

Reviewer #3:

Remarks to the Author:

In this resubmission of the manuscript looking at clinical immune responses to different ATL-DC-based regimens in glioma patients, the authors have been very responsive to all of the comments made from each of the reviewers.

In particular, they have effectively softened the interpretation overall with inclusion of important caveats in key places when discussing the results and findings of these analyses. They have also effectively clarified the rationale for reporting the results at this point in the study with the numbers of patients, and that this is a precursor to subsequent studies to further evaluate and confirm these findings. Finally, it is appreciated the additional modeling that was used to help account for important potential sources of bias such as latent confounders and inpatient variability. They have put in a lot of work and the result is a very nice and improved paper.

While overall very responsive to the comments, there are only a few minor issues that still need to be fixed for the final manuscript:

1. For comment #3d, the authors have generated Kaplan-Meier (KM) estimates for the OS and TTP for the different groups; however, this really should replace the values that are currently included in Table 1 or these OS and TTP estimates should be removed altogether from a table focused on baseline characteristics. Thus, I would recommend simply removing OS and TTP from Table 1 altogether, but if it really is of interest to include this here, then the KM estimates (medians and corresponding 95% confidence intervals) should be used instead in this table. This should be an easy change.

In Table 1, the KM estimates, along with their corresponding medians and 95% confidence intervals, are now listed.

2. While caveats have effectively been included, lines 282-284 should be softened as well. the analyses here are not confirming significant predictors, but show a potentially useful predictor even when adjusting for other factors. This can be confirmed in larger subsequent studies.

We have further updated the verbiage on this particular statement to include the final statement, "This can be confirmed in larger subsequent studies."

3. In the statistical methods lines 575-576, the OS and TTP distributions were *compared* using the log-rank test and graphical evaluation of these curves were assessed using the methods of Kaplan and Meier. The survminer R package is just applying this methodology, and it can be referenced parenthetically that it was used, but the methods were those of Kaplan and Meier. This is again an easy fix and just improves the specific accuracy in describing the methodology.

We have amended the statistical methods to include this exact language.